# Probiotic Supplementation Enhances the Effects of a Nutritional Intervention on Quality of Life in Women with Hashimoto’s Thyroiditis—A Double-Blind Randomised Study

**DOI:** 10.3390/nu17213387

**Published:** 2025-10-28

**Authors:** Karolina Osowiecka, Damian Skrypnik, Joanna Myszkowska-Ryciak

**Affiliations:** 1Department of Dietetics, Institute of Human Nutrition Sciences, Warsaw University of Life Sciences (WULS), 02-776 Warsaw, Poland; 2Department of Family Medicine, Poznan University of Medical Sciences, 60-355 Poznan, Poland

**Keywords:** hypothyroidism, diet, nutritional intervention, probiotic, female

## Abstract

**Background**: Hashimoto’s thyroiditis (HT) is characterised by chronic inflammation of the thyroid gland. The impact of a health-promoting diet and probiotics on health and quality of life, as well as on the anti-thyroid peroxidase antibody (anti-TPO), is increasingly being researched. However, the relevance of these factors to the course of HT is yet to be fully established. **Objective**: The aim of this study was to assess the impact of a 12-week nutritional intervention, comprising a rational, health-promoting diet supplemented with the probiotic strain *Lactiplantibacillus plantarum 299v* (*Lp299v*), on eating habits, nutritional status, health and quality of life in patients diagnosed with HT. **Methods**: The 12-week study involved 64 female patients with HT, divided into two groups: the NE+*Lp299v* group, which received nutritional education and *Lp299v* (n = 32); and the NE+placebo group, which received nutritional education and placebo (n = 32). Before and after the intervention, anthropometric parameters, body composition analysis, blood pressure, blood anti-TPO levels, dietary habits, quality of life, and gastrointestinal symptoms were assessed. **Results**: The NE+*Lp299v* intervention improved overall quality of life (60.94 pts. vs. 35.94 pts.), including 12 of 14 domains, and the diet quality index (11.03 pts. vs. 18.50 pts.). The NE+placebo group improved overall quality of life (54.69 pts. vs. 39.84 pts.), including 3 of 14 domains, and the diet quality index (12.34 pts. vs. 19.18 pts.). Anti-TPO blood levels and body mass index did not improve in either group. **Conclusions**: *Lp299v* can enhance the efficacy of nutritional education in improving the quality of life of individuals diagnosed with HT. However, these benefits appear to be independent of anti-TPO levels.

## 1. Introduction

Hashimoto’s thyroiditis (HT) is one of the most common autoimmune diseases that gradually destroys the thyroid parenchyma and negatively affects the quality of life in patients [1]. The disease manifests more frequently in women than in men (4:1 ratio) [2]. It is not fully understood why women are more often affected than men, although the cause is believed to be the role of female sex hormones or X chromosome inactivation [3]. The diagnosis is made by the physician following the identification of elevated levels of antibodies against thyroid peroxidase (anti-TPO) and thyroglobulin (anti-TG). The presence of anti-TPO can be detected in 95% of patients, while anti-TG is observed in 60–80% of cases [1,3]. Also important in diagnosis are the occurrence of clinical symptoms [1,3] and an ultrasound examination, which can observe the infiltration of T and B lymphocytes into the thyroid tissues [3]. The infiltration leads to the atrophy of thyroid cells, which results in the development of subclinical or overt hypothyroidism [4]. Inflammation in HT has also been demonstrated to have a detrimental effect on quality of life [5,6], particularly in the neuropsychiatric domain [6,7]. The treatment regimen involves the administration of levothyroxine (LT4), which improves hormone levels in hypothyroidism coexisting with HT. The treatment of Hashimoto’s disease represents a significant financial burden to both patients and the state. According to the National Health Fund, over PLN 114 million was spent on levothyroxine treatment in Poland in 2023; of this amount, the state reimbursed less than PLN 62 million [8].

It is increasingly recognised that diet is a necessary element of HT treatment. Previous studies, with varying results, have examined elimination diets (e.g., gluten-free, lactose-free, iodine-restricted, or restricted to selected foods), energy deficits, and the supplementation of black cumin [9,10]. The following dietary regimens have been employed: an autoimmune protocol [11,12], a low-carb diet [13], and a Mediterranean diet [10,14,15]. The anti-inflammatory nature of the Mediterranean diet, which is comparable to a rational, balanced diet, may result in a reduction in inflammation. Consequently, the Mediterranean diet could be considered a dedicated dietary strategy in HT [10,14,16]. Such a diet contains many food products with health-promoting effects and is therefore varied and provides nutrients with an anti-inflammatory effect on anti-TPO and/or anti-TG antibodies. These nutrients include, but are not limited to, vitamin D [17,18], selenium [19], iron [20,21,22], vitamin C [23], magnesium, zinc [24], iodine [25], and eicosapentaenoic and docosahexaenoic acids [26]. Nevertheless, given the variety of intervention strategies and their duration, as well as the differing group sizes, it is still challenging to determine the most suitable diet for HT [9].

The impact of microbiota and probiotics on autoimmune diseases, including HT, is a subject that is being discussed with increasing frequency. In a ten-week supplementation study, the administration of a synbiotic containing *Lactobacillus casei*, *Lactobacillus acidophilus*, *Lactobacillus rhamnosus*, *Lactobacillus bulgaricus*, *Bifidobacterium breve*, *Bifidobacterium longum*, *Streptococcus thermophilus*, and fructo-oligosaccharide resulted in improvements in blood pressure and quality of life in patients diagnosed with hypothyroidism. However, the study observed no significant impact on depression and TSH levels [27]. AkbariRad et al. [28] reported beneficial effects for synbiotics (a blend of bacteria, including *Lactobacillus rhamnosus*, *Lactobacillus plantarum*, *Lactobacillus casei*, *Lactobacillus helveticus*, *Lactobacillus acidophilus*, *Lactobacillus bulgaricus*, *Lactobacillus gasseri*, *Bifidobacterium bifidum, Bifidobacterium lactis*, *Bifidobacterium breve*, *Bifidobacterium longum*, *and Streptococcus thermophilus*, along with fructooligosaccharides) supplementation on fatigue and thyroid function in hypothyroid patients. However, two meta-analyses examining the effect of probiotics and/or prebiotics on the course of various thyroid diseases concluded that probiotics did not significantly improve anti-TPO or anti-TG activity in HT [29,30].

To the best of our knowledge, no study yet has been conducted with the *Lactiplantibacillus plantarum 299v* strain (*Lp299v*) among patients with HT. A growing body of research is indicating its immunological effects, e.g., it improves inflammation in studies on animal models of inflammatory bowel disease [31], and in humans it reduces the concentration of IL-8 and IL-12 [32,33]. *Lp299v* may improve iron metabolism [34,35], the course of irritable bowel syndrome [36], and cognitive function in patients with depression [37]. Therefore, it may be posited that *Lp299v* could have therapeutic use in patients with HT as an adjunct to pharmacological and dietary treatment.

The objective of the present study was to evaluate the impact of a 12-week nutritional intervention, characterised by an anti-inflammatory, Mediterranean-style diet, augmented by the probiotic strain *Lp299v*, on the dietary habits, nutritional status, health, and quality of life of female patients diagnosed with HT. It is hypothesised that *Lp299v*, when incorporated into a balanced, healthy diet, may yield additional health benefits, particularly in terms of quality of life. In order to enhance the study’s validity, a randomised, double-blind design was employed.

## 2. Materials and Methods

### 2.1. Study Design and Ethical Approval

The study design comprised 12 weeks of individualised nutritional education, provided by a dietitian, and the administration of the probiotic *Lp299v* (the “NE+*Lp299v*” or probiotic group) or a placebo (the “NE+placebo” or placebo group). The study was a single-centre, double-blind dietary intervention in which the probiotic and placebo capsules appeared identical. All measurements were collected during visits to the Dietetics Clinic at the Department of Dietetics, Warsaw University of Life Sciences (SGGW). The data collection process was executed through the utilisation of CAWI (Computer-Assisted Online Interviewing) and PAPI (Paper-and-Pen-In-Person Interviewing) methodologies. Nutrition education was conducted either online or at the Dietetics Clinic, depending on the participant’s preferences or capabilities. The personal results obtained were provided to participants free of charge. Prior to the commencement of the study, participants were furnished with comprehensive information regarding its objectives, scope, and the option of withdrawing at any stage without the need to provide a justification.

The study was conducted in accordance with the Declaration of Helsinki and approved by the Ethics Committee of the Institute of Human Nutrition, Warsaw University of Life Sciences WULS, Poland (Resolutions No. 22/2021, 18 June 2021 and No. 21/2022, 18 July 2022) for studies involving humans. Participation in the study was voluntary, and informed consent was obtained from all subjects. The participants did not receive any form of financial compensation or reimbursement for their involvement in the study.

### 2.2. Participant Selection

Recruitment of the study sample was based on inclusion and exclusion criteria, conducted using snowball sampling, using social media platforms, primarily social media groups for residents of Warsaw and the surrounding area and dedicated to Hashimoto’s thyroiditis. The study was also promoted at a partner medical facility and pharmacy.

The following inclusion criteria were employed: (1) a diagnosis of Hashimoto’s thyroiditis based on the presence of anti-TPO and anti-TG antibodies, or characteristic ultrasound images indicative of HT; (2) thyroid function assessed for euthyroidism based on the results of tests performed by the patient; (3) female gender; (4) age between 18 and 64 years; (5) normal body weight (BMI 18.50–24.99 kg/m^2^) or excessive body weight (BMI ≥ 25.00 kg/m^2^); and/or a low-quality diet (low pHDI-10) [38]; (6) informed consent to participate in the study. The following criteria were used to determine exclusion from the study: (1) thyroid diseases other than HT, cancer, coeliac disease, Dühring’s disease or gluten allergy; (2) pregnancy, lactation; (3) administration of *Lp299v* or weight loss medications.

To calculate the minimum number of participants in both groups, a 50% improvement in overall quality of life after the intervention was presumed (assuming a 5% significance level and 80% power). Based on this, each group should have 47 participants. A minimum of 100 patients was assumed to be eligible for the study, to account for potential dropout.

### 2.3. Randomisation and Blinding

Patients were randomly assigned to a probiotic group (nutrition education with *Lp299v*) or a placebo group (nutrition education with placebo) according to a computer-generated randomisation list created by an investigator not involved in the study. The study was double-blind, meaning that neither the patient nor the investigator conducting the experiment knew which group had received the probiotic until the end of the nutritional intervention.

### 2.4. Intervention Procedure

A 12-week individual nutritional education programme was conducted in two groups. Additionally, one group received the *Lp299v* strain (NE+*Lp299v* group) and the other received a placebo (NE+P group). Prior to and after the education programme, dietary habits were evaluated using a three-day food and beverage intake questionnaire. Patients were also asked to complete questionnaires to assess: (1) dietary habits (FFQ-6 and KomPAN^®^ questionnaire [38,39]); (2) quality of life with thyroid disease (ThyPROpl) [40]; (3) the Gastrointestinal Symptom Rating Scale (GSRS) [41]. Anthropometric measurements (body weight and height, waist and hip circumference), body composition analysis, peripheral blood pressure, and measurement of anti-TPO antibody concentration in blood were also performed. All parameters were measured at the baseline and after a 12-week period of intervention in order to assess the difference, and if possible, to compare to reference values. The study design is presented below (Figure 1).

### 2.5. Nutrient Intake and Diet Quality

The estimation of energy and nutrient intake at the baseline and after the intervention was assessed using a three-day food and beverage intake questionnaire recommended by the Polish Society of Dietetics and the National Consultant for Family Medicine [42]. Patients were asked to record all food and drink consumed, including spices, particularly salt, for two typical days, e.g., weekdays, and one non-typical day (e.g., weekends). The estimation of energy and nutrient intake was assessed using the “DietetykPro^®^” dietary programme (Wrocław, Poland). The programme incorporates a Polish nutrient database for food products and dishes [43], and the United States Department of Agriculture (USDA) nutrient database [44]. In instances where food items were not present in the database, the most similar product was selected for the purpose of nutrient calculation. Nutrient intake was compared with Polish dietary standards and recommendations from 2020 [45]. Energy intake was compared to the individual’s total energy requirement, which was estimated by multiplying their basal metabolic rate, calculated using the Mifflin-St-Jeor equation [46], by their physical activity index (PAL) based on the interview: 1.4–1.69—for a sedentary or lightly active lifestyle; 1.7–1.99—for an active or moderately active lifestyle; and 2.00–2.40—for a very active lifestyle [47].

The Polish-validated Food Frequency Questionnaire (FFQ-6) was used to assess the frequency of consumption of selected food products and dietary habits. Participants were asked to select one of six categories indicating the frequency of food consumption in the past 12 months: (1) never or almost never (0 times/day); (2) once a month or less often (0.025 times/day); (3) several times a month (0.1 times/day); (4) several times a week (0.571 times/day); (5) daily (1 time/day); (6) several times a day (2 times/day) [39].

The quality of the diet was evaluated using two validated questionnaires: the Healthy Diet Index (pHDI-10) and the Unhealthy Diet Index (nHDI-14) [38]. The pHDI-10 contains 10 questions pertaining to the frequency of consuming health-promoting foods, while the nHDI-14 comprises 14 questions regarding the frequency of consumption unhealthy foods. The cafeteria of answers, with designated daily frequencies assigned to each option were as follows: (1) never (0 times/day); (2) 1–3 times a month (0.06 times/day); (3) once a week (0.14 times/day); (4) few times a week (0.5 times/day); (5) once a day (1 time/day); (6) few times a day (2 times/day). The ranges for both the pHDI-10 and nHDI-14 were as follows: scores of 0–33 points indicate a low index; 34–66 points indicate a moderate index, and 67–100 points indicate a high index. The DQI (Diet-Quality Index) was also estimated, with the calculation based on the sum of the positive pHDI-10 and negative nHDI-14 scores. The DQI scale used was as follows: (1) −100 to −26 points–a high intensity of non-healthy dietary characteristics; (2) −25 to 25 points-a low intensity of non-healthy and pro-healthy dietary characteristics; (3) 26 to 100 points-a high intensity of pro-healthy dietary characteristics [38].

### 2.6. Selected Lifestyle and Health Factors

All participants were questioned about their medical care, medications, dietary supplements, and herbal remedies. Patients were requested to refrain from modifying their supplementation regimen (with the exception of vitamin D supplementation) and to report any alterations to their medication.

### 2.7. Quality of Life

The quality of life of study participants at the baseline and after the intervention was assessed using two instruments: (1) the Thyroid Disease Quality of Life Questionnaire (ThyPROpl) [40]; and (2) the Gastrointestinal Symptom Rating Scale (GSRS) [41]. ThyPROpl is a thyroid-specific questionnaire validated for the Polish population by Sawicka-Gutaj et al. [40]. It captures both physical and psychological dimensions of disease impact, including symptoms, emotional functioning, cognitive complaints, fatigue, and social participation. The questionnaire consists of a 13-point scale including questions regarding: (1) goitrogen symptoms; (2) hyperthyroid symptoms; (3) hypothyroid symptoms; (4) eye symptoms; (5) tiredness; (6) cognitive complaints; (7) anxiety; (8) depressivity; (9) emotional susceptibility; (10) impaired social life; (11) impaired daily life; (12) impaired sexual life; (13) cosmetic complaints; and (14) overall quality of life. The ThyPROpl questionnaire uses a 5-point Likert scale, where respondents indicate the frequency or intensity of specific symptoms and experiences (e.g., from “not at all” to “very much”) [40]. The score range is 0 to 100, with higher scores reflecting greater symptom burden or lower quality of life.

The validated Polish version of the GSRS consists of 15 test items grouped into five symptom groups (reflux, abdominal pain, indigestion, diarrhoea, and constipation). The response scale is 7-point, where 1 indicates no bothersome symptoms and 7 indicates very bothersome symptoms [41].

### 2.8. Nutritional and Health Status

All anthropometric measurements were performed in accordance with the Anthropometry Procedures Manual by a trained dietitian using standardised equipment [48]. Body weight (BW, kg) was measured to the nearest 0.1 kg using an electronic digital scale (ACCUNIQ BC720, SELVAS Healthcare, Korea), with participants wearing light indoor clothing and no shoes. Height (H, cm) was assessed in the Frankfort horizontal plane, barefoot, with a precision of 0.1 cm. Waist circumference (WC, cm) and hip circumference (HC, cm) were measured with an inelastic measuring tape (SECA 201, Hamburg, Germany) to the nearest 0.1 cm, following standardised procedures [49]. Central obesity was defined as waist-to-hip ratio (WHR) > 0.8 [50] and/or waist-to-height ratio (WHtR) ≥ 0.5 [51]. Overweight was defined as a body mass index (BMI) between 25.00 and 29.99 kg/m^2^, while obesity was defined as a BMI ≥ 30.00 kg/m^2^ [52]. Body composition (body fat, muscle mass, and total body water) was analysed with the bioelectrical impedance (BIA) method using the body composition analyser (ACCUNIQ BC-720, SELVAS Healthcare, Korea). This BIA method was omitted in patients with contraindications to the procedure, including those with metal parts in their bodies. Detailed measurement procedures are described in the previously published protocol (Section 3.8.) [53].

Peripheral blood pressure was examined with the subject in a sitting position, after a minimum of 5 min of rest, with the ACCUNIQ BC-250. Detailed measurement procedures are described in the previously published protocol (Section 3.9.) [53].

The anti-TPO titter was assessed in venous blood samples obtained from a cubital vein of the forearm by qualified medical laboratory personnel. Serum concentrations were measured using immunochemistry on the “Alinity I” analyser (Abbott Laboratories, Chicago, IL, USA) and compared with the reference value of <5.61 IU/mL. Detailed measurement procedures are described in the previously published protocol (Section 3.10.) [53].

### 2.9. Nutrition Education

The nutritional education programme (6 individual meetings spaced about 2 weeks apart, a total of 6 h of education) is described in detail in the protocol in Section 3.11. [53]. All educational materials were based on scientific literature and standards of the Polish Society of Dietetics and the National Institute of Public Health–National Institute of Hygiene. Each patient received not only educational materials, but also a sample menu based on individual energy needs, illustrating appropriate meals and portion sizes. Throughout the intervention, patients were provided with an access to a dietitian support, and this was not limited to meetings.

### 2.10. Lactiplantibacillus plantarum 299v

The experimental group received 1 × 10^10^ CFU (colony-forming units) of *Lp299v* contained in Sanprobi IBS^®^. The carrier was potato starch and magnesium salts of fatty acids (capsule shell: hydroxypropyl methylcellulose). The control group received a placebo capsule containing granulated potato starch and magnesium stearate (capsule shell: hydroxypropyl methylcellulose). Patients were instructed to take one capsule (probiotic in the experimental group and placebo capsule in the control group) once daily with a meal. Side effects of *Lactiplantibacillus plantarum 299v* are rare and do not pose a threat to the patient. Detailed information regarding the Lp299v capsules and placebo intake can be found in the protocol in Section 3.12. [53].

### 2.11. Statistical Analyses

All statistical analyses were performed using Statistica 13.1 PL (StatSoft Inc., Tulsa, OK, USA; StatSoft, Krakow, Poland). Descriptive statistics were generated for all variables of interest. Normality of the variables was assessed using the Shapiro–Wilk test before statistical analysis. The student *t*-test was used for quantitative data with a normal distribution, and the Mann–Whitney test for non-normally distributed data. The Wilcoxon test was used for ordinal dependent variables within groups (before vs. after education). The two-way repeated measures ANOVA with post hoc tests was performed to assess the effectiveness of the nutritional intervention and to observe differences between the probiotic and placebo groups. A Bonferroni post hoc analysis was then used to determine which group showed significance. The differences in categorical variables between groups were tested using the Pearson chi-square test, and the McNemar test was used for comparisons within the group (before vs. after intervention). The average treatment effect (ATE) was calculated as the mean difference in the obtained value between the probiotic group and the placebo group. A *p* value of ≤0.05 was considered as statistically significant.

## 3. Results

### 3.1. Characteristics of the Study Group

A total of 64 women with HT completed the study, 32 of whom were assigned to the probiotic group (NE+*Lp299v*), and the remaining 32 were assigned to the placebo group (NE+P). All participants had euthyroid thyroid function. The average age in the probiotic group was 41.53 ± 11.77 years, and in the placebo group, 39.28 ± 8.97 years. Levothyroxine administration was reported by 84% of the probiotic group and 78% of the placebo group. Neither age nor the levothyroxine usage differed between the groups (Table 1).

### 3.2. Effect of the Intervention on Nutritional Status and Health Parameters

Prior to the intervention, the mean BMI in both groups indicated that the subjects were overweight. The mean WHR and WHtR scores indicated the risk of metabolic diseases, respectively. The mean blood pressure scores indicated no hypertension among studied individuals. The probiotic group showed a decrease in hip circumference, systolic blood pressure, and diastolic blood pressure after the intervention. However, the nutritional intervention with *Lp299v* or placebo had no effect on other anthropometric measurements, body composition, pulse rate, and the anti-PRO concentration. The magnitude of change over time showed a significant reduction in systolic blood pressure in the probiotic group. However, in the case of WHR and muscle mass, the placebo group demonstrated favourable outcomes (Table 2).

The body weight status and other health risk indicators examined, based on the waist circumference, WHR, and WHtR (e.g., metabolic risk, abdominal obesity), did not differentiate between groups, either before or after the intervention. The intervention, in the form of nutritional education with the administration of probiotics or a placebo, resulted in a higher percentage of people with normal body weight in the probiotic group and a lower percentage of people with “no abdominal obesity” according to the WHR index (Appendix A: Appendix A).

### 3.3. Effect of the Intervention on Quality of Life

Prior to the intervention, subjects in both groups reported fatigue as the most bothersome symptom, followed closely by decreased overall quality of life and depression. Following the intervention, fatigue was reported as the most bothersome symptom, with depression ranking second. In the placebo group, the overall quality of life dropped to third place, and in the probiotic group, it dropped to fourth place. The least bothersome symptom prior to the intervention in the probiotic group was impaired social life. In contrast, the least bothersome symptom in the placebo group was goitre symptoms (Table 3). The implemented nutritional intervention with probiotics or placebo significantly improved quality of life measured by the ThyPROpl questionnaire over time in almost all aspects of life, except for goitrogenic symptoms. Subsequent post hoc analyses indicated more favourable outcomes in the probiotic group in comparison to the placebo group (significant change for 12 of the 14 categories vs. 3 of the 14 categories, respectively). The ATE value indicated that the addition of probiotics to nutritional education improved the women’s quality of life in all areas except hyperthyroid symptoms. In the probiotic group, the overall quality of life score decreased significantly from baseline (mean difference = −25.0 points, 95% CI: −42.3 to −7.7; t(31) = −2.95; *p* = 0.006), with an estimated statistical power of approximately 85%. In the placebo group, the overall quality of life score also decreased significantly (mean difference = −14.9 points, 95% CI: −29.2 to −0.5; t(31) = −2.11; *p* = 0.043), although the statistical power of this comparison was limited (≈57%).

With regard to the GSRS, the most troublesome reported symptom in both groups was indigestion, both prior to and following the intervention. The least problematic symptom in all individuals was found to be reflux (and diarrhoea in the probiotic group prior to the intervention). The nutritional intervention was found to have a beneficial effect on all symptoms. In post hoc analyses, a beneficial effect was detected in three out of five domains, while for the placebo group, this effect was observed in two out of five domains, respectively. The ATE value indicated that the addition of probiotics to nutritional education improved the symptoms in three domains: constipation, abdominal pain and reflux (Table 3).

### 3.4. Effect of the Nutritional Education on Diet Quality and Dietary Intake

Examined diet quality index categories (pHDI-10, nHDI-14, DQI) did not differ between the groups before and after the intervention. Following the intervention, there was a significant decrease in the frequency of low-intensity unhealthy traits and a significant increase in the frequency of high-intensity health-promoting traits in the placebo group (Table 4). Higher percentage of better dietary quality according to the DQI and moderate pHDI levels were also observed in the placebo and probiotic groups. (Appendix A). The intervention had a significant impact on dietary quality indicators, with positive scores increasing and negative scores decreasing. Post hoc analyses confirmed a significant effect of time for both groups on the DQI and nHDI-14 (Table 4).

Nutrition education increased the frequency of consumption of some of the healthy foods listed in the pHDI-10 index, including whole-wheat bread in the probiotic group and buckwheat, oatmeal and whole-grain pasta in the placebo group. In both groups, the frequency of fish consumption showed an increase following the intervention (Appendix A, Appendix A). Consequently, the educational programme led to a decline in the frequency of consumption of selected products from the n-HDI-14 index. These products were: fast food, cheese and alcohol in the probiotic group; fried foods, butter and sweetened beverages in the placebo group, and cold cuts, sausages and sweets in both groups. The frequency of tomato consumption increased significantly in the probiotic group after the education programme.

Following the nutritional education programme, several beneficial changes were observed in energy and nutrient intake. The intake of energy, carbohydrates, sucrose, total fat, and saturated fat decreased, while the intake of alpha-linolenic acid, omega 3, and vitamin D increased. In post hoc analyses, the significant effect remained for the probiotic group for the majority of the analysed components. In this group, the beneficial effect of education was found to be significantly greater for EPA+DHA and vitamin B_12_ (Table 5).

However, nutrition education did not result in an increase in the number of people meeting dietary guidelines for vitamins and minerals (with the exception of vitamin B_12_ in the placebo group). In this context, no significant differences were observed between the two groups, either before or after the educational intervention (Appendix A, Appendix A).

## 4. Discussion

Nutrition education has been shown to improve diet quality, which can in turn enhance various aspects of quality of life. Furthermore, it has been observed to reduce symptoms of hyperthyroidism, fatigue, and abdominal pain. The *Lp299v* supplementation enhanced the improvement in quality of life across almost all domains. The probiotic group also experienced greater improvements in blood pressure (systolic and diastolic) and hip circumference. However, the study found that nutrition education with probiotics or placebo did not reduce anti-TPO titres, BMI, waist circumference, or body composition.

In our study, the mean BMI remained unchanged, although the percentage of individuals with a normal body weight increased. Other studies have observed a reduction in BMI following a Mediterranean diet [14,54] or a combined diet (Mediterranean + gluten-free) [14]. Regarding other parametric and body composition characteristics, our study observed a reduction in hip circumference but also a higher percentage of individuals with abdominal obesity according to WHR. In the Ülker study, the Mediterranean diet and the Mediterranean + gluten-free diet demonstrated reduced waist and hip circumference [14]. Probiotics may also affect obesity by modifying the gut microbiota, enriching and increasing its diversity. This modification also reduces inflammation and oxidative stress, as well as endotoxemia, which is associated with the development of obesity [55]. Regarding the effect of probiotics on body weight, one candidate may be *Akkermansia muciniphila.* According to a meta-analysis conducted by Liu et al., involving animal studies, *A. muciniphila* significantly reduced body weight gain by approximately 10% [55,56]. The number of studies examining the effect of *Lp299v* on obesity is limited. In a human study, a probiotic mixture (*Lp299v* + *Saccharomyces cerevisiae* var. *boulardii*) with octacosanol did not change BMI [57].

The nutritional education programme incorporating probiotics resulted in a marked improvement in blood pressure (systolic and diastolic) compared to the education programme with a placebo. It is well-documented that dietary patterns have a significant impact on blood pressure, with certain diets, notably the Mediterranean diet, having demonstrated both preventive and therapeutic benefits [58,59,60]. However, the effect of probiotics in this regard remains unclear. In a study with Lp299v supplementation, no reduction in blood pressure was observed [32], while in another study, systolic blood pressure increased [33] after the use of this probiotic. However, the study by Naruszewicz demonstrated a decrease in systolic blood pressure in patients supplementing with Lp299v [61].

In the study, anti-TPO levels were reduced, though not to a statistically significant degree, in both groups. In another 8-week study, a synbiotic taken by patients with HT also did not affect anti-TPO levels [62]. As demonstrated by Piticchio’s meta-analysis [63], a gluten-free diet, which is a diet that has been extensively studied in the context of Hashimoto’s disease, has been shown to reduce anti-TPO levels. The Mediterranean diet, when followed for 12 weeks, has been shown to have a beneficial therapeutic effect on anti-TPO levels [14] or to have no effect [10]. The combination of a gluten-free and Mediterranean diet has been shown to bring a measurable benefit in anti-TPO concentration [14]. Levels of anti-TPO decreased after six months on a low-energy diet with selenium and zinc supplementation, with or without the elimination of selected products [64]. A similar effect was also obtained on another low-calorie diet with selenium and zinc supplementation [65] and on a low-carbohydrate diet (12–15% carbohydrates, 50–60% protein, 25–30% fat) [13,65]. A 12-week diet in compliance with the autoimmune protocol resulted in a negligible increase in anti-TPO concentration [11]. According to the literature, diets high in sugar, refined carbohydrates, salt, sodium or trans fatty acids have been shown to promote pro-inflammatory effects. Limiting the intake of the aforementioned foods and balancing the diet with health-promoting foods, such as fruits, vegetables, lean meat, olive oil, fish, and whole grains, ensures the provision of nutrients (omega-3 fatty acids, zinc, magnesium, iron, iodine, selenium, protein, and vitamin B_12_), thus promoting anti-inflammatory effects and better thyroid function [20,66,67,68,69,70,71,72,73]. It is particularly recommended to supplement with vitamin D, which effectively reduces anti-TPO titres if supplementation lasts at least 3 months [17,18,74].

In our study, the most noticeable symptoms included fatigue, depression, indigestion, and a general decrease in quality of life. This finding aligns with other studies, which identified fatigue as one of the most bothersome symptoms [7,75]. There is a possibility that it is correlated with anti-TPO and anti-TG antibodies, as observed by Li et al. [7]. Treatment with LT4 for hypothyroidism improved quality of life, although 15% of patients still reported persistent symptoms [76]. Another review revealed that, despite achieving euthyroidism with LT4, 5–10% of patients reported an unsatisfactory quality of life, mood disturbances, and symptoms of hypothyroidism [77]. The risk of a depression diagnosis in patients with hypothyroidism in autoimmune thyroiditis is 3.5 times higher than in individuals without the condition [78]. In our study, improved cognitive function was observed in the education group with probiotics. Similar results were obtained in another study, where *Lp299v* improved concentration in individuals with severe depression [37]. Hypothyroidism is frequently associated with indigestion and a range of other gastrointestinal disorders [79]. The *Lp299v* strain has a documented effect on gastrointestinal health and reduces the symptoms of irritable bowel syndrome [80]. Apart from the effect of *Lp299v* on gastrointestinal symptoms, there is little research on its effect on other aspects of quality of life [31]. A growing body of research has consistently demonstrated the positive impact of probiotics on various aspects of quality of life. Studies have reported improvements in patients with irritable bowel syndrome [81], dyspepsia [82], and sexual function in women, particularly those receiving antidepressant treatment [83]. Additionally, probiotics have been shown to enhance the quality of life in various autoimmune diseases, such as systemic sclerosis, psoriasis, and coeliac disease. In a study on patients with hypothyroidism, a synbiotic (containing strains of the genera *Lactobacillus*, *Bifidobacterium*, and *Streptococcus thermophilus*) positively affected physical pain, mental health, general health perception, and vitality, but not depressive symptoms or physical functioning [27].

A diet rich in nutrients is important for perceived health [84,85,86]. With regard to the impact of diet on quality of life in cases of Hashimoto’s disease, we identified one study [87]. In this study, the most common complaints reported by patients with HT were fatigue, drowsiness, poor concentration, and dry skin. After 12 weeks of autoimmune protocol use, the number of people suffering from these symptoms decreased [11]. Among other autoimmune diseases, an anti-inflammatory diet has been shown to reduce symptoms associated with rheumatoid arthritis, including pain [87,88], while the Mediterranean and ketogenic diets have been found to minimise symptoms in multiple sclerosis [89]. There is also a study in which patients with HT experienced improvement in depressive symptoms after supplementation, particularly with vitamin D, but also with selenomethionine or myoinositol [79]. To summarise, both *Lp299v* and diet have been shown to enhance various aspects of quality of life. This may be due to modulating the gut microbiota and reducing intestinal dysbiosis [90], which occurs in people with HT [91]. Dysbiosis is particularly important in gastrointestinal symptoms [90] and neuropsychiatric disorders [92].

This probiotic/placebo nutritional education programme has several strengths. Our study has added to our understanding of the impact of diet and probiotic therapy on the course and quality of life of HT. The 12-week study of nutritional education and dietary changes was sufficient to produce measurable benefits in many aspects of quality of life. Another key strength of the programme is its educational approach, which goes beyond mere menu recommendations to provide more comprehensive dietary guidance. This educational component was found to be highly beneficial for patients, enhancing their ability to sustain healthy eating habits beyond the study’s conclusion. In addition, there was ongoing contact with a dietitian and consultations every two weeks, which allowed for gradual changes, as well as support and knowledge transfer from a specialist. Another key strength of the study is the examination of the effect of *Lp299v*, a potentially anti-inflammatory antibody, on the immune function of HT, which is the anti-TPO antibody.

The study also has several limitations. Firstly, the sample selection method was both purposeful and random. Secondly, the accuracy of nutrient estimates derived from the Food Register Questionnaire is contingent on the patient’s diligence and precision in providing their dietary information. Nevertheless, in order to minimise this error, all participants were provided with comprehensive instructions on how to complete the questionnaires and were able to contact the researcher at any stage. Determining nutritional biomarkers, e.g., vitamins in blood serum, would constitute an objective verification of self-assessment. However, such tests have certain limitations in terms of their use, for example, when assessing the consumption of whole grains, fruit and vegetable diversity, or legume consumption (the overall diet quality). Another confounding factor is the inclusion of individuals with both normal and excessive body weight, which may have contributed to the lack of changes in body weight in the groups during the intervention. Varying disease duration and the use (or non-use) of levothyroxine may also be confounding factors. Another limitation of the study is the limited external validity and generalisability of the findings to the broader population with HT due to the method of patient recruitment and the inclusion and exclusion criteria. The lack of inclusion of individuals with hypothyroidism may also have been a confounding factor, contributing to the lack of significance in the quality-of-life domains of goitrogen symptoms and hypothyroid symptoms. Further research is also needed to verify the impact of the placebo effect, mainly on the quality of life in people with HT. Another limitation of the study is the sample size and statistical power for the NE+placebo group in the case of overall quality of life. Finally, financial constraints limited the assessment of blood parameters to anti-TPO antibodies, precluding a more comprehensive analysis of immune and metabolic markers. This may weaken the interpretation of the mechanisms underlying the effects of *Lp299v*.

## 5. Conclusions

Nutritional education combined with the probiotic *Lactiplantibacillus plantarum 299v* has been shown to improve overall quality of life (including tiredness, depressivity, cosmetic complaints, and constipation) and blood pressure in patients suffering from Hashimoto’s thyroiditis. The study found no impact on nutritional status or blood anti-TPO levels. However, the results should be treated with caution due to the limitations of the study. It is possible that a longer intervention is required or that education was not as effective in this regard compared to other diet-based interventions. Future research should employ larger, randomised trials with extended follow-up to clarify the long-term impact of integrated nutritional and probiotic interventions in Hashimoto’s thyroiditis. A longer-term follow-up, for example, at 6 or 12 months post-intervention, would undoubtedly be beneficial in evaluating the sustainability of dietary changes and quality-of-life improvements observed in this study.

## Figures and Tables

**Figure 1 nutrients-17-03387-f001:**
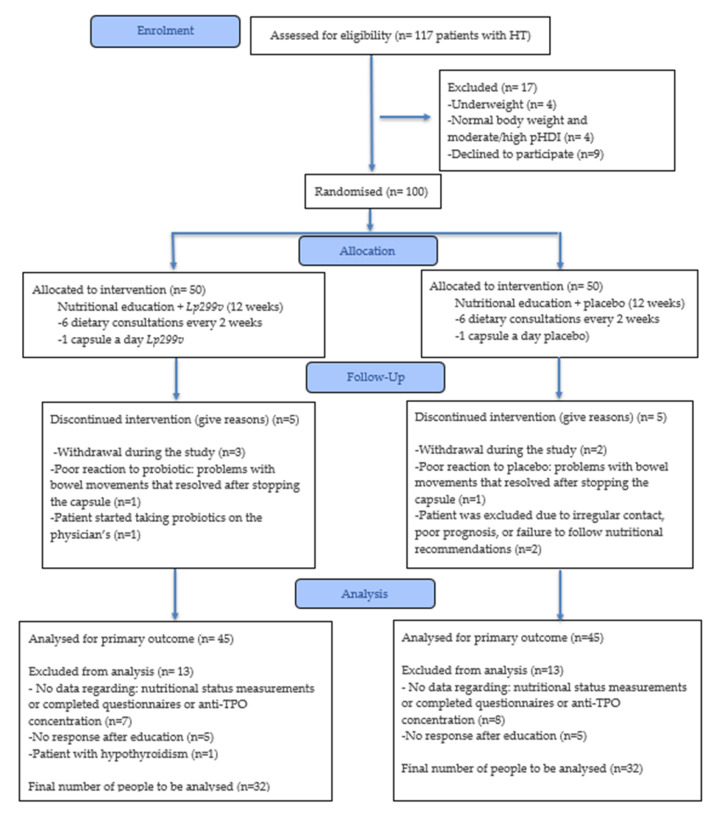
The study diagram. HT—Hashimoto’s thyroiditis; pHDI-10—Pro-Healthy Diet Index 10; NE+*Lp299v*—nutrition education + *Lactiplantibacillus plantarum 299v*; NE+P—nutrition education + placebo; anti-TPO-antibodies against thyroid peroxidase.

**Table 1 nutrients-17-03387-t001:** Group characteristics (SD, standard deviation; Me, median; Min—minimum, Max—maximum).

Parameters	NE+*Lp299v* Group (n = 32)	NE+P Group (n = 32)	*p*-Value
Age [years]	41.53 ± 11.77	39.28 ± 8.97	0.393 †
Mean ± SD (Me; Min–Max)	(41.00; 22–64)	(39.50; 22–54)	
Levothyroxine administration N (%)			0.286 ‡
No	5 (15.63)	7 (21.88)	
Yes	27 (84.38)	25 (78.12)	

†—the student *t*-test; ‡—the chi-square test.

**Table 2 nutrients-17-03387-t002:** Effect of the intervention on nutritional status and health parameters among the studied women. Results are presented as mean ± standard deviation.

Parameters	NE+*Lp299v* Group (n = 32)		NE+P Group (n = 32)		ANOVA *p*-Value			Magnitude of Change ∆		*p*-Value
	Baseline	Post Intervention	Baseline	Post Intervention	Group Effect	Time Effect	Time × Group Effect	NE+*Lp299v*	NE+P	
BMI [kg/m^2^]	28.8 ± 4.69	28.8 ± 4.87	26.9 ± 4.65	27.0± 4.64	0.115	0.980	0.688	−0.05 ± 1.20	0.05 ± 0.75	0.804 ^†^
Waist circumstance [cm]	95.1 ± 13.36	95.0 ± 12.98	90.7 ± 12.18	89.7 ± 11.66	0.124	0.261	0.292	−0.44 ± 3.40	−0.95 ± 2.77	0.509^‡^
Hip circumstance [cm]	109.8 ± 8.83 ^A^	107.5 ± 8.78 ^B^	106.2 ± 8.15	105.6 ± 8.36	0.192	0.009	0.089	−2.25 ± 3.23	−0.5 ± 4.74	0.167 ^†^
WHR	0.86 ± 0.08	0.88 ± 0.08 *	0.85 ± 0.08	0.80 ± 0.06 *	0.205	0.147	0.008	0.02 ± 0.03	−0.01 ± 0.04	0.030 ^†^
WHtR	0.57 ± 0.08	0.57 ± 0.08	0.54 ± 0.08	0.50 ± 0.07	0.096	0.289	0.294	0.0 ± 0.02	−0.01 ± 0.02	0.294 ^‡^
Fat mass [kg] ^&^	29.7 ± 9.34	29.9 ± 9.78	26.4 ± 9.45	26.2 ± 9.13	0.156	0.942	0.332	0.28 ± 2.25	−0.24 ± 1.76	0.331 ^‡^
Fat-free mass [kg] ^&^	48.8 ± 6.06	48.6 ± 6.42	48.8 ± 4.99	49.1 ± 5.10	0.849	0.714	0.273	−0.15 ± 1.39	0.31 ± 1.78	0.476 ^†^
Muscle mass [kg] ^&^	27.2 ± 3.41	27.1 ± 3.57	27.2 ± 2.80	27.4 ± 2.83	0.857	0.746	0.332	−0.08 ± 0.76	0.16 ± 1.02	0.050 ^†^
TBW [Lt.] ^&^	35.7 ± 4.46	35.6 ± 4.67	35.7 ± 3.67	35.9 ± 3.68	0.849	0.760	0.349	−0.10 ± 1.01	0.20 ± 1.34	0.062 ^†^
Systolic pressure [mmHg]	129.4 ± 14.05 ^a,A,^*	121.1 ± 14.42 ^B^	118.4 ± 12.13 ^b^	117.5 ± 14.24 *	0.019	0.006	0.029	−8.33 ± 13.45	−0.95 ± 12.86	0.028 ^‡^
Diastolic pressure [mmHg]	76.2 ± 9.69 ^A^	71.0 ± 8.29 ^B^	69.8 ± 10.02	68.9 ± 11.26	0.053	0.015	0.077	−5.22 ± 8.88	−0.86 ± 10.44	0.077 ^‡^
Pulse [bpm]	71.59 ± 11.19	71.89 ± 10.49	70.47 ± 8.72	75.7 ± 14.08	0.573	0.065	0.099	0.30 ± 8.85	5.27 ± 14.26	0.124 ^†^
Anti-TPO [IU/mL]	453.1 ± 525	418.4 ± 467.32	318.9 ± 443.58	278.8 ± 411.01	0.235	0.065	0.892	−34.64 ± 203.31	−40.07 ± 97.14	0.416 ^†^

Note: The two-way repeated measures ANOVA, statistically significant differences with the post hoc Bonferroni test (*p* ≤ 0.05) are indicated as follows: ^a,b^—for the group effect, ^A,B^—for the time effect, and *—for the time and group effect; ∆—delta represents the change in the variable during intervention; ^†^—the U Mann–Whitney test; ^‡^—the student *t*-test; ^&^—probiotic group n = 28, control group n = 30; BMI—the body mass index; WHR—the waist-to-hip ratio; WHtR—the waist-to-height ratio; TBW—total body water; anti-TPO—antibodies against thyroid peroxidase.

**Table 3 nutrients-17-03387-t003:** Effect of the intervention on quality of life among the studied women. Results are presented as mean and standard deviation.

Parameters	NE+*Lp299v* Group (n = 32)		NE+P Group (n = 32)		ANOVA *p*-Value			ATE
	Baseline	Post Intervention	Baseline	Post Intervention	Group Effect	Time Effect	Time × Group Effect	
ThyPROpl								
Goitrogen symptoms	24.79 ± 20.22	18.18 ± 17.08	13.85 ± 13.97	13.99 ± 13.42	0.045	0.075	0.063	−6.75
Hyperthyroid symptoms	36.23 ± 25.88 ^A^	26.17 ± 21.12 ^B^	33.20 ± 17.04 ^A^	21.48 ± 17.09 ^B^	0.416	<0.001	0.692	1.66
Hypothyroid symptoms	48.83 ± 25.57	37.89 ± 23.17	43.95 ± 22.98	37.30 ± 18.61	0.577	0.004	0.463	−4.30
Eye symptoms	33.89 ± 24.21 ^A^	25.00 ± 18.98 ^B^	24.32 ± 18.71	18.26 ± 15.74	0.070	0.001	0.511	−2.83
Tiredness	67.97 ± 21.09 ^A^	54.13 ± 22.75 ^B^	71.65 ± 14.82 ^A^	58.37 ± 18.43 ^B^	0.351	<0.001	0.910	−0.56
Cognitive complaints	42.45 ± 26.04 ^A^	29.30 ± 22.29 ^B^	36.59 ± 22.34	30.60 ± 21.21	0.660	<0.001	0.167	−7.16
Anxiety	41.15 ± 23.37 ^A^	25.65 ± 19.32 ^B^	41.41 ± 18.96	34.64 ± 20.94	0.314	<0.001	0.081	−8.72
Depressivity	51.90 ± 24.19 ^A^	39.17 ± 21.01 ^B^	45.54 ± 22.28	41.18 ± 21.12	0.649	0.004	0.148	−8.37
Emotional susceptibility	48.70 ± 23.35 ^A^	34.38 ± 16.68 ^B^	45.31 ± 21.88	36.72 ± 17.69	0.904	<0.001	0.268	−5.73
Impaired social life	23.83 ± 19.66 ^A^	13.28 ± 18.63 ^B^	20.51 ± 17.98	16.41 ± 18.83	0.981	0.002	0.149	−6.44
Impaired daily life	32.66 ± 26.28 ^A^	16.72 ± 24.32 ^B^	24.35 ± 18.09	15.70 ± 19.31	0.338	<0.001	0.192	−7.29
Impaired sex life	36.72 ± 36.33 ^A^	19.92 ± 28.19 ^B^	35.16 ± 32.13	34.38 ± 34.64	0.387	0.018	0.031	−16.01
Cosmetic complaints	45.18 ± 27.98 ^A^	26.95 ± 23.23 ^B^	36.20 ± 21.20	28.13 ± 23.50	0.462	<0.001	0.087	−10.16
Overall quality of life	60.94 ± 32.96 ^A^	35.94 ± 34.74 ^B^	54.69 ± 26.52 ^A^	39.84 ± 29.69 ^B^	0.867	<0.001	0.146	−10.16
GSRS								
Diarrhoea	2.06 ± 1.52	1.93 ± 1.35	2.49 ± 1.52	1.91 ± 1.21	0.505	0.049	0.215	0.45
Indigestion	3.23 ± 1.49	2.67 ± 1.10	3.32 ± 1.13 ^A^	2.52 ± 1.10 ^B^	0.907	<0.001	0.426	0.23
Constipation	2.89 ± 1.79 ^A^	2.11 ± 1.29 ^B^	2.75 ± 1.64	2.06 ± 1.52	0.785	<0.001	0.830	−0.08
Abdominal pain	2.78 ± 1.31 ^A^	2.08 ± 0.92 ^B^	2.75 ± 0.97 ^A^	2.13 ± 1.13 ^B^	0.982	<0.001	0.795	−0.07
Reflux	2.06 ± 1.34 ^A^	1.45 ± 0.86 ^B^	2.03 ± 1.16	1.64 ± 0.94	0.730	0.002	0.482	−0.25

Note: The two-way repeated measures ANOVA, statistically significant differences with the post hoc Bonferroni test (*p* ≤ 0.05) are indicated as follows: ^A,B^—for the time effect; ATE—the Average Treatment Effect; ThyPROpl—the thyroid-specific questionnaire; GSRS—the Gastrointestinal Symptom Rating Sc.

**Table 4 nutrients-17-03387-t004:** Effect of the nutritional education on diet quality among the studied women. Results are presented as mean and standard deviation.

Parameters	NE+*Lp299v* Group (n = 32)		NE+P Group (n = 32)		ANOVA *p*-Value			Magnitude of the Change ∆		*p*-Value
	Baseline	Post Intervention	Baseline	Post Intervention	Group Effect	Time Effect	Group × Time Effect	NE+*Lp299v* Group	NE+P Group	
pHDI-10	25.47 ± 11.49	28.43 ± 10.06	26.19 ± 8.51	28.07 ± 9.83	0.933	0.050	0.656	2.96 ± 8.04	1.88 ± 11.09	0.656 ‡
nHDI-14	14.44 ± 6.74 ^A^	9.93 ± 5.00 ^B^	13.85 ± 7.71 ^A^	8.89 ± 5.28 ^B^	0.558	<0.001	0.764	−4.50 ± 5.88	−4.96 ± 6.20	0.930 †
DQI	11.03 ± 11.92 ^A^	18.50 ± 11.05 ^B^	12.34 ± 9.58 ^A^	19.18 ± 9.39 ^B^	0.649	<0.001	0.832	7.47 ± 10.18	6.84 ± 13.28	0.930 †

Note: The two-way repeated measures ANOVA, statistically significant differences with the post hoc Bonferroni test (*p* ≤ 0.05) are indicated as follows: ^A,B^—for the time effect; ∆—delta represents the change in the variable during intervention; †—the U Mann–Whitney test; ‡—the student *t*-test; pHDI—the Prohealthy-Diet-Index; nHDI—the Non-Healthy Diet Index; DQI—the Diet Quality Index.

**Table 5 nutrients-17-03387-t005:** Effect of the nutritional education on dietary intake among the studied women (n = 49). Results are presented as mean and standard deviation.

Parameters	NE+*Lp299v* Group (n = 26)		NE+P Group (n = 23)		ANOVA *p*-Value			Magnitude of the Change ∆		*p*-Value
	Baseline	Post Intervention	Baseline	Post Intervention	Group Effect	Time Effect	Group × Time Effect	NE+*Lp299v* Group	NE+P Group	
Energy [kcal]	1990.2 ± 456.64 ^A^	1723.5 ± 322.55 ^B^	1964.5 ± 333.27	1789.6 ± 226.44	0.812	<0.001	0.391	−266.7 ± 413.24	−174.9 ± 314.09	0.602 †
Protein [g]	85.4 ± 19.70	87.4 ± 15.40	86.8 ± 17.43	82.8 ± 14.33	0.679	0.720	0.318	1.9 ± 20.10	−4.0 ± 21.16	0.318 ‡
Fat [g]	78.7 ± 26.10	69.0 ± 19.18	79.2 ± 17.63	74.0 ± 16.76	0.596	0.014	0.451	−9.6 ± 23.26	−5.2 ± 16.07	0.451 ‡
Carbohydrates [g]	240.4 ± 62.51 ^A^	195.6 ± 47.46 ^B^	231.0 ± 61.61	207.4 ± 40.56	0.928	<0.001	0.239	−44.8 ± 64.55	−23.6 ± 58.91	0.249 †
Dietary fibre [g]	25.0 ± 8.39	22.8 ± 6.59	23.0 ± 6.85	23.0 ± 7.75	0.619	0.363	0.350	−2.2 ± 9.45	0.0 ± 6.38	0.350 ‡
Sucrose [g]	34.1 ± 22.08	22.4 ± 16.00	34.1 ± 17.81	24.7 ± 13.17	0.773	0.001	0.717	−11.7 ± 22.68	−9.4 ± 20.51	0.928 †
SFA [g]	26.6 ± 10.65 ^A^	21.5 ± 8.37 ^B^	26.5 ± 7.30	23.6 ± 7.01	0.632	0.003	0.394	−3.4 ± 11.75	−1.7 ± 7.25	0.552 ‡
MUFA [g]	30.5 ± 11.62	27.1 ± 9.81	30.4 ± 9.63	28.6 ± 8.13	0.783	0.078	0.552	−3.4 ± 11.75	−1.7 ± 7.25	0.568 †
EPA + DHA [mg]	275.4 ± 347.58	607.3 ± 749.23	612.6 ± 852.81	442.6 ± 681.68	0.572	0.509	0.045	331.9 ± 763.13	−170.0 ± 938.47	0.045 ‡
Linoleic acid [g]	10.2 ± 4.38	11.4 ± 5.02	10.4 ± 3.94	11.2 ± 4.52	0.974	0.285	0.777	1.2 ± 7.13	0.7 ± 5.39	0.992 †
α-linolenic acid [g]	1.4 ± 0.80	1.9 ± 1.23	1.8 ± 0.73	2.3 ± 1.19	0.087	0.011	0.869	0.5 ± 1.36	0.5 ± 1.35	0.960 †
Total n-3 FA [g]	1.7 ± 1.09 ^A^	2.6 ± 1.53 ^B^	2.5 ± 1.06	2.8 ± 1.57	0.125	0.008	0.258	0.89 ± 1.71	0.4 ± 1.37	0.258 ‡
Total n-6 FA [g]	9.2 ± 3.65	9.1 ± 2.93	10.0 ± 3.73	9.7 ± 3.61	0.352	0.788	0.879	−0.1 ± 4.56	−0.3 ± 4.88	0.879 ‡
PUFA [g]	11.9 ± 4.85	13.5 ± 4.04	13.2 ± 4.30	14.3 ± 6.05	0.344	0.109	0.772	1.6 ± 5.45	1.1 ± 6.13	0.772 ‡
Sodium [mg]	2599.3 ± 1091.86	2559.3 ± 1251.55	2716.4 ± 1429.6	2441.6 ± 510.99	0.999	0.369	0.502	−40.0 ± 1039.52	−274.8 ± 1384.45	0.880 †
Calcium [mg]	788.8 ± 229.17	748.4 ± 280.84	820.8 ± 231.80	908.9 ± 397.59	0.148	0.645	0.218	−40.4 ± 06.88	88.1 ± 411.23	0.258 †
Magnesium [mg]	381.0 ± 91.38	366.1 ± 91.51	358.8 ± 96.06	364.8 ± 86.31	0.588	0.766	0.485	−14.9 ± 135.14	6.0 ± 47.84	0.485 ‡
Iron [mg]	15.0 ± 3.18	14.0 ± 3.69	14.4 ± 3.38	13.3 ± 4.47	0.409	0.092	0.974	−1.0 ± 5.12	−1.1 ± 3.14	0.974 ‡
Zinc [mg]	11.3 ± 2.61	10.6 ± 2.50	10.9 ± 3.23	10.5 ± 1.99	0.687	0.194	0.639	−0.7 ± 3.49	−0.4 ± 2.16	0.465 †
Iodine [µg]	68.5 ± 37.44	68.1 ± 28.03	61.7 ± 34.54	70.8 ± 27.39	0.789	0.418	0.373	−0.4 ± 41.59	9.2 ± 31.71	0.373 ‡
Vit. D (diet + supplementation) [µg]	39.6 ± 41.65	52.6 ± 39.72	55.1 ± 51.87	70.4 ± 45.68	0.153	**0.016**	0.834	13.0 ± 31.7	15.3 ± 46.79	0.373 †
Folate [µg]	382.3 ± 151.25	370.9 ± 94.45	400.5 ± 113.19	362.5 ± 87.89	0.852	0.217	0.503	−11.3 ± 160.66	−38.0 ± 105.79	0.489 †
Vit. B_12_ [µg]	3.7 ± 1.66	4.3 ± 2.34	4.9 ± 3.27	4.0 ± 1.97	0.476	0.611	**0.048**	0.5 ± 2.51	−0.9 ± 2.25	**0.039** †

Note: The two-way repeated measures ANOVA, statistically significant differences with the Post Hoc Bonferroni test (*p* ≤ 0.05) are indicated as follows: ^A,B^—for the time effect; ∆—delta represents the change in the variable during intervention; †—the U Mann–Whitney test; ‡—the student *t*-test; EPA—eicosapentaenoic acid; DHA—docosahexaenoic acid; FA—fatty acids; SFA—saturated fatty acids; MUFA—monounsaturated fatty acids; PUFA—polyunsaturated fatty acids.

## Data Availability

The data presented in this study are available on request from the corresponding author. The data are not publicly available due to internal regulations.

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
