# Peer review of "Probiotic Supplementation Enhances the Effects of a Nutritional Intervention on Quality of Life in Women with Hashimoto’s Thyroiditis—A Double-Blind Randomised Study"

_nutrients, 2025, doi:10.3390/nu17213387_

Round 1
Reviewer 1 Report
Comments and Suggestions for Authors
This study presents a well-structured investigation into the adjunctive role of Lactiplantibacillus plantarum 299v (Lp299v) in enhancing the outcomes of nutritional education for patients with Hashimoto’s thyroiditis (HT). Its thorough documentation of dietary adherence, quality-of-life metrics, and immunological markers provides a comprehensive overview of the intervention’s clinical relevance. However, several points merit further consideration:
1. The study primarily emphasizes improvements in quality of life and dietary habits. Could the authors further elaborate on the therapeutic thresholds and safety profile of Lp299v, particularly in the context of long-term use and potential interactions with thyroid hormone replacement therapy?
2. The paper suggests that Lp299v contributes to improved quality-of-life outcomes independent of anti-TPO levels. Could further exploration and validation of this hypothesis through additional immunological markers or thyroid function tests (e.g., TSH, free T4) strengthen the conclusions regarding the probiotic’s mechanism of action?
3. While the intervention duration was 12 weeks, the sustainability of dietary changes and quality-of-life improvements remains unclear. Would a follow-up assessment at 6 or 12 months post-intervention provide insight into the durability and long-term impact of the combined approach?
4. The manuscript reports improvements in 12 of 14 quality-of-life domains in the NE+Lp299v group. Could the authors clarify which domains did not improve and discuss potential reasons or confounding factors that may have influenced these outcomes?
5. The study mentions gastrointestinal symptom assessment but does not elaborate on specific findings. Could inclusion of these results, particularly in relation to probiotic efficacy, provide a more holistic view of patient-reported outcomes? Given that Lp299v is known to influence gut health, detailing changes in gastrointestinal symptoms—such as bloating, bowel regularity, or abdominal discomfort—could substantiate its clinical relevance and enhance the interpretability of quality-of-life improvements. Moreover, correlating these symptom changes with dietary adherence and microbiota modulation may offer valuable insights into the mechanistic underpinnings of the intervention’s benefits.
6. Finally, the discussion could benefit from a more critical reflection on the limitations of self-reported dietary assessments and the potential influence of placebo effects in quality-of-life evaluations. How might future studies incorporate objective dietary biomarkers or blinded assessments to mitigate these concerns?
Author Response
We are very grateful for all remarks, comments and suggestions to our manuscript. Please find below the Authors’ responses to each of the Reviewer comments. Changes made to the manuscript are marked in gray. We hope it will make reading them easier.
Kind regards,
Authors
Reviewer comments:
- The study primarily emphasizes improvements in quality of life and dietary habits. Could the authors further elaborate on the therapeutic thresholds and safety profile of Lp299v, particularly in the context of long-term use and potential interactions with thyroid hormone replacement therapy?
Authors:
Thank you for your comment. Indeed, our main goal was to assess quality of life and dietary habits of women with HT. According to a study conducted by the Quality Institute of the Jagiellonian Center of Innovation, a minimum of 106 CFU of Lp299v per day is required to give a therapeutic effect. This dose was sufficient to achieve a therapeutic effect in patients with cardiovascular diseases, which we quoted in our article (insert citations).
Regarding the safety issues, the trials by the Quality Institute of the Jagiellonian Center of Innovation and the Institute of Children's Health Center (Poland) showed that the preparation is safe for consumers. The preparation has been registered by the Chief Sanitary Inspectorate in Poland and acknowledged to be safe to be sold without physician prescription (a dose of max. 2 capsules/day is recommended). We mention this in our protocol: https://www.mdpi.com/2075-4426/13/12/1659, which we cited in the present manuscript. Additionally, the Quality Institute of the Jagiellonian Center of Innovation conducted a 3-month study indicating a constant amount of this strain in the preparation. To the best of our knowledge, there are no known interactions between Lp299v and Levothyroxine (LT4). Our patients took Lp299v with a meal. Eating the first meal should be separated from LT4, to allow proper absorption of the medicine. Despite the lack of knowledge about the interaction between Lp299v and LT4, the time interval was maintained.
Reviewer comments:
- The paper suggests that Lp299vcontributes to improved quality-of-life outcomes independent of anti-TPO levels. Could further exploration and validation of this hypothesis through additional immunological markers or thyroid function tests (e.g., TSH, free T4) strengthen the conclusions regarding the probiotic’s mechanism of action?
Authors:
Thank you very much for this insightful suggestion. We agree that further exploration and validation of the proposed hypothesis through additional immunological markers and thyroid function tests (e.g., TSH, free T4) could indeed strengthen the understanding of the probiotic’s mechanism of action. Lp299v exhibits certain anti-inflammatory and immune-modulating properties, so it is plausible that it might influence immunological markers in Hashimoto’s disease. However, the overall role of Lp299v in immunological processes remains poorly understood, which makes it difficult to draw definitive conclusions at this stage. It is also worth noting that the literature indicates that normal thyroid function does not necessarily correlate with a good quality of life. In our study, all participants had normal thyroid function yet experienced improvements following the intervention. Such effects could potentially also occur in individuals with hypothyroidism, although further studies would be required to confirm this.
We fully acknowledge the importance of including a broader panel of blood parameters, such as anti-TG antibodies and thyroid hormones, in future investigations. Unfortunately, financial constraints of our project limited us to measuring anti-TPO antibodies, which we considered the most relevant indicator of thyroid autoimmunity, as they are elevated in a higher proportion of Hashimoto’s thyroiditis cases compared to anti-TG.
We hope that our findings will encourage further research into this area and more comprehensive assessments of the immunological mechanisms underlying the observed effects. As noted in the revised manuscript, we have explicitly mentioned this limitation (highlighted in text):
“Finally, financial constraints limited the assessment of blood parameters to anti-TPO antibodies, precluding a more comprehensive analysis of immune and metabolic markers. This may weaken the interpretation of the mechanisms underlying the effects of Lp299v.”
Reviewer comments:
- While the intervention duration was 12 weeks, the sustainability of dietary changes and quality-of-life improvements remains unclear. Would a follow-up assessment at 6 or 12 months post-intervention provide insight into the durability and long-term impact of the combined approach?
Authors:
Thank you for your comment. We do believe that a follow-up evaluation would be the most helpful in assessing the effectiveness of the intervention in terms of dietary changes and quality of life. We have included this statement in our conclusions:
„A longer-term follow-up, for example at 6 or 12 months post-intervention, would undoubtedly be beneficial in evaluating the sustainability of dietary changes and quality-of-life improvements observed in this study.”
Reviewer comments:
- The manuscript reports improvements in 12 of 14 quality-of-life domains in the NE+Lp299vgroup. Could the authors clarify which domains did not improve and discuss potential reasons or confounding factors that may have influenced these outcomes?
Authors:
The domains "goitroge symptoms" and "hypothyroid symptoms" did not improve. This may be due to the exclusion of individuals with hypothyroidism. We added this confounder to the study limitations. We have added the following sentence to the study limitations:
“The lack of inclusion of individuals with hypothyroidism may also have been a confounding factor, contributing to the lack of significance in the quality of life domains “goitrogeous symptoms” and “hypothyroid symptoms.”
Reviewer comments:
- The study mentions gastrointestinal symptom assessment but does not elaborate on specific findings. Could inclusion of these results, particularly in relation to probiotic efficacy, provide a more holistic view of patient-reported outcomes? Given that Lp299vis known to influence gut health, detailing changes in gastrointestinal symptoms—such as bloating, bowel regularity, or abdominal discomfort—could substantiate its clinical relevance and enhance the interpretability of quality-of-life improvements. Moreover, correlating these symptom changes with dietary adherence and microbiota modulation may offer valuable insights into the mechanistic underpinnings of the intervention’s benefits.
Authors:
Thank you very much for your suggestion. We've added the sentence in the discussion:
„To summarise, both Lp299v and diet have been shown to enhance various aspects of quality of life. This may be due to modulating the gut microbiota and reducing intestinal dysbiosis [90], which occurs in people with HT [91]. Dysbiosis is particularly important in gastrointestinal symptoms [90] and neuropsychiatric disorders [92].”
Reviewer comments:
- Finally, the discussion could benefit from a more critical reflection on the limitations of self-reported dietary assessments and the potential influence of placebo effects in quality-of-life evaluations. How might future studies incorporate objective dietary biomarkers or blinded assessments to mitigate these concerns?
Authors:
Thank you for your question. We tried to ensure that the results of the self-assessment of diet were as reliable as possible. To this end, we used both available questionnaires assessing eating habits and a food diary. Data from the questionnaire and the food diary were consistent. Another step was to explain how to complete these questionnaires and what to remember, such as entering salt, beverages, etc. We also provided a sense of security and comfort to increase the chances of honest responses. We realize that despite our and the patients' best efforts, dietary data do not fully reflect their actual diet, how many nutrients they consumed through their diet, etc. Blood tests would certainly be a better biomarker of nutrient intake adequacy; however, such tests are limited. It is difficult to assess, for example, dietary patterns, intake of vegetables, legumes or whole grain products. We included the self-assessment of diet in the study limitations. We also included placebo effects, which could have had a psychobiological impact on the study results, in the study limitations. The risk of concerns about blinded assessment is difficult to eliminate. In our study, to reduce this risk, we used a double-blind design. The capsule packaging looked identical in both groups (placebo vs. probiotic). Certainly, more research in the future could allay these concerns.
The sentence regarding the placebo effect is:
“Further research is also needed to verify the impact of the placebo effect, mainly on the quality of life in people with HT.”
The sentence regarding the dietary biomarkers is:
“Determining nutritional biomarkers, e.g. vitamins in blood serum, would constitute an objective verification of self-assessment. However, such tests have certain limitations in terms of their use, for example when assessing the consumption of whole grains, fruit and vegetable diversity, or legume consumption (the overall diet quality).”
Reviewer 2 Report
Comments and Suggestions for Authors
This randomized double-blind study investigated the impact of a 12-week nutritional education program combined with the probiotic Lactiplantibacillus plantarum 299v (Lp299v) on 64 women with Hashimoto’s thyroiditis. Participants were assigned to receive either dietary counseling with probiotics or with placebo. Both groups showed improvements in diet quality and quality of life, but the probiotic group reported broader and stronger improvements across most quality-of-life domains. No significant changes were observed in anti-thyroid peroxidase antibody levels or BMI, indicating that the benefits were largely independent of immunological parameters.
Issues to address:
-Underpowered sample size: Power analysis indicated ~47 participants per group were required (94 total), but only 64 completed the study (32/group). This reduction likely compromises the ability to detect true between-group effects, especially for subtle changes.
-Recruitment bias: Participants were recruited via social media and self-selection, which introduces volunteer bias (more health-conscious individuals). This limits external validity and generalizability to the broader Hashimoto’s population.
-Heterogeneity of participants: Inclusion of both normal-weight and overweight individuals, as well as those with varying disease duration and levothyroxine use, introduces confounding variability. Stratified analysis by BMI or medication status would strengthen conclusions.
-Short intervention period (12 weeks): The timeframe may be too short to detect meaningful changes in anti-TPO titers, body composition, or inflammation-related outcomes. Hashimoto’s thyroiditis has a slow immunological course; a longer follow-up (≥6 months) would be more informative.
-Single biomarker (anti-TPO): No measurement of TSH, free T4/T3, anti-TG, CRP, cytokines, or gut microbiota composition has been performed, which weakens mechanistic interpretation of probiotic effects.
-Dietary assessment limitations: The use of 3-day dietary records and self-reported FFQs is vulnerable to recall bias and under/overestimation. Objective dietary biomarkers (e.g., plasma fatty acids, serum carotenoids) could validate self-reports.
Overall, although the study is interesting, several methodological and statistical issues limit its robustness. The study presents interesting preliminary findings, but conclusions about the probiotic’s efficacy should be interpreted cautiously. Therefore, a major revision should address sample size justification, statistical correction, and potential biases.
Author Response
We are very grateful for all remarks, comments and suggestions to our manuscript. Please find below the Authors’ responses to each of the Reviewer comments. Changes made to the manuscript are marked in yellow. We hope it will make reading them easier.
Kind regards,
Authors
Reviewer comments:
-Underpowered sample size: Power analysis indicated ~47 participants per group were required (94 total), but only 64 completed the study (32/group). This reduction likely compromises the ability to detect true between-group effects, especially for subtle changes.
Authors:
Thank you for your comment. We previously included the statistical power for both groups on lines 337-342 (in results):
“In the probiotic group, the overall quality of life score decreased significantly from baseline (mean difference = −25.0 points, 95% CI: −42.3 to −7.7; t(31) = −2.95; p = 0.006), with an estimated statistical power of approximately 85%. In the placebo group, the overall quality of life score also decreased significantly (mean difference = −14.9 points, 95% CI: −29.2 to −0.5; t(31) = −2.11; p = 0.043), although the statistical power of this comparison was limited (≈57%).“
We added the information about the insufficient sample size to the study limitations:
“Another limitation of the study is the sample size and statistical power for the NE+placebo group in the case of overall quality of life.”
Reviewer comments:
-Recruitment bias: Participants were recruited via social media and self-selection, which introduces volunteer bias (more health-conscious individuals). This limits external validity and generalizability to the broader Hashimoto’s population.
Authors:
Thank you for your comment. In our group we included only individuals who had a poor diet quality and/or were overweight/obese, but not only by social media but also with the help of clinic. Indeed, participation in the study was voluntary so we can assume that individuals might have been more health-conscious but they were lacking knowledge and support. Based on the literature, we believe that people who take care of their health, including a healthy diet, proper body weight, etc., generally might have a good quality of life, so intervention in such individuals might not yield measurable results.
The added sentence reads:
“Another limitation of the study is the limited external validity and generalizability of the findings to the broader population with HT due to the method of patient recruitment and the inclusion and exclusion criteria.”
Reviewer comments:
-Heterogeneity of participants: Inclusion of both normal-weight and overweight individuals, as well as those with varying disease duration and levothyroxine use, introduces confounding variability. Stratified analysis by BMI or medication status would strengthen conclusions.
Authors:
Thank you for your comment. In our study, it would be difficult to divide the groups according to the above-mentioned divisions due to the probiotic/placebo division. Even if we were to divide the study into four groups (i.e., a study group with a normal BMI, a study group with an excessive BMI, a control group with a normal BMI, and a control group with an excessive BMI), the number of participants in these groups might be significantly too small. We have already included the issue of different BMI levels in the study limitations:
Another confounding factor is the inclusion of individuals with both normal and excessive body weight, which may have contributed to the lack of changes in body weight in the groups during the intervention.”
So we added a sentence about the duration of the disease and taking LT4
„Varying disease duration and the use (or non-use) of levothyroxine may also be confounding factors.”
Reviewer comments:
-Short intervention period (12 weeks): The timeframe may be too short to detect meaningful changes in anti-TPO titers, body composition, or inflammation-related outcomes. Hashimoto’s thyroiditis has a slow immunological course; a longer follow-up (≥6 months) would be more informative.
Authors:
Thank you for your comment. In this case, a longer intervention might have been more effective, although 12 weeks was sufficient to improve dietary quality and quality of life. 12 weeks is also a period sufficient to observe body weight and body composition changes. In several other studies, a 12-week intervention also produced measurable benefits, for example, in anti-TPO antibody titers: https://pubmed.ncbi.nlm.nih.gov/38370054/ , https://www.mdpi.com/2072-6643/13/3/862 . The anti-inflammatory effect of Lp299v on cytokines was also observed after 6 weeks of intervention: https://pubmed.ncbi.nlm.nih.gov/33597583/ , https://pubmed.ncbi.nlm.nih.gov/30355158/ . Therefore, we wanted to see if a shorter intervention would produce the expected results. A shorter intervention also made it easier to maintain patient motivation to remain in the study. I In our conclusions, we suggested that researchers conduct a longer intervention in the future:
„It is possible that a longer intervention is required, or that education was not as effective in this regard compared to other diet-based interventions. Future research should employ larger, randomized trials with extended follow-up to clarify the long-term impact of integrated nutritional and probiotic interventions in Hashimoto’s thyroiditis.”
Reviewer comments:
-Single biomarker (anti-TPO): No measurement of TSH, free T4/T3, anti-TG, CRP, cytokines, or gut microbiota composition has been performed, which weakens mechanistic interpretation of probiotic effects.
Authors:
Thank you for your comment. We planned to include not only anti-TPO but also anti-TG and hormones assessing thyroid function as part of the blood test. Unfortunately, financial constraints prevented us from doing so. Therefore, we limited ourselves to the most important indicator (anti-TPO), which indicates inflammation, and a higher percentage of patients with HT have elevated levels of it (than anti-TG). However, we hope that our results will encourage other researchers to conduct further studies in this area. We have already included this limitation of the study in the following sentence:
“Finally, financial constraints limited the assessment of blood parameters to anti-TPO antibodies, precluding a more comprehensive analysis of immune and metabolic markers.”
However, we decided to add an additional sentence regarding this limitation. The fragment now reads as follows:
“Finally, financial constraints limited the assessment of blood parameters to anti-TPO antibodies, precluding a more comprehensive analysis of immune and metabolic markers. In the future, it would be worthwhile to consider objective markers that would allow for the assessment of nutritional status for selected nutrients, such as DHA, EPA, or iodine.”
Reviewer comments:
-Dietary assessment limitations: The use of 3-day dietary records and self-reported FFQs is vulnerable to recall bias and under/overestimation. Objective dietary biomarkers (e.g., plasma fatty acids, serum carotenoids) could validate self-reports.
Authors:
We agree with this comment. Data regarding patient diets is always subject to some degree of error. Objective markers that would allow for the assessment of nutritional status for selected nutrients, such as vitamins, DHA, EPA, or iodine would be beneficial to collect more objective data. However, such tests have certain limitations in terms of their use in dietary assessment, for example when assessing the consumption of whole grains, fruit and vegetable diversity, or legume consumption. so in the limitations of the study we wrote earlier that:
„Secondly, the accuracy of nutrient estimates derived from the Food Register Questionnaire is contingent on the patient's diligence and precision in providing their dietary information. Nevertheless, in order to minimise this error, all participants were provided with comprehensive instructions on how to complete the questionnaires and were able to contact the researcher at any stage.”
We've now added a supplementary sentence to make this restriction more explicit (Your comment is consistent with another reviewer's comment, please look for the corrected sentence in grey). It reads:
“Determining nutritional biomarkers, e.g. vitamins in blood serum, would constitute an objective verification of self-assessment. However, such tests have certain limitations in terms of their use, for example when assessing the consumption of whole grains, fruit and vegetable diversity, or legume consumption (the overall diet quality).”
Reviewer 3 Report
Comments and Suggestions for Authors
This is a very well written MS that covers a very topical theme.
Some minor points:
- Figure 1 legend: it should read "Prisma diagram",
- The authors need to be a bit more critical about the value of BMI today. It's well documented that BMI is not a true reflection of the obese/over-weighted people,
- I would invite the authors to elaborate more the conclusions part of the MS.
Author Response
We are very grateful for all remarks, comments and suggestions to our manuscript. Please find below the Authors’ responses to each of the Reviewer comments. Changes made to the manuscript are marked in green. We hope it will make reading them easier.
Kind regards,
Authors
Reviewer comments:
- Figure 1 legend: it should read "Prisma diagram",
Authors:
Thank you for your comment. We believe the original name of the figure is more appropriate. Prisma stands for "Preferred Reporting Items for Systematic Reviews and Meta-Analyses," so it refers to a different type of publication than ours.
Reviewer comments:
- The authors need to be a bit more critical about the value of BMI today. It's well documented that BMI is not a true reflection of the obese/over-weighted people,
Authors:
Thank you for your comment. In our study we did not rely solely on BMI. To comprehensively assess the nutritional status of the patients, we also assessed other indicators, such as WHR, WHtR, metabolic risk based on waist circumference, and body composition analysis.
Reviewer comments:
- I would invite the authors to elaborate more the conclusions part of the MS.
Authors:
- Was:
„Nutritional education combined with the probiotic Lactiplantibacillus plantarum 299v has been shown to improve quality of life and blood pressure in patients suffering from Hashimoto's thyroiditis. The study found no impact on nutritional status or blood anti-TPO levels. It is possible that a longer intervention is required, or that education was not as effective in this regard compared to other diet-based interventions. Future research should employ larger, randomized trials with extended follow-up to clarify the long-term impact of integrated nutritional and probiotic interventions in Hashimoto’s thyroiditis.”
- Current version:
„Nutritional education combined with the probiotic Lactiplantibacillus plantarum 299v has been shown to improve overall quality of life (including tiredness, depressivity, cosmetic complaints, constipation) and blood pressure in patients suffering from Hashimoto's thyroiditis. The study found no impact on nutritional status or blood anti-TPO levels. However, the results should be treated with caution due to the limitations of the study. It is possible that a longer intervention is required, or that education was not as effective in this regard compared to other diet-based interventions. Future research should employ larger, randomized trials with extended follow-up to clarify the long-term impact of integrated nutritional and probiotic interventions in Hashimoto’s thyroiditis.”
Round 2
Reviewer 2 Report
Comments and Suggestions for Authors
ISSUES HAVE BEEN ACKNOWLEDGED, ALTHOGH LIMITATIONS IN THE SCIENTIFIC ROBUSTNESS OF THE DATA STILL PERSIST